# Continuous Ropivacaine Peroneal Nerve Infiltration for Fibula Free Flap in Cervicofacial Cancer Surgery: A Randomized Controlled Study

**DOI:** 10.3390/jcm11216384

**Published:** 2022-10-28

**Authors:** Cyrus Motamed, Frederic Plantevin, Jean Xavier Mazoit, Morbize Julieron, Jean Louis Bourgain, Valerie Billard

**Affiliations:** 1Service d’Anesthésie, Gustave Roussy Cancer Center, 94800 Villejuif, France; 2Laboratoire d’anesthésie, Paris-Saclay University, INSERM U1195 Faculté de Médecine de Bicêtre 63 Rue Gabriel Péri, 94270 Le Kremlin-Bicêtre, France; 3Service de Chirurgie Cervico Faciale, Gustave Roussy Cancer Center, 94800 Villejuif, France

**Keywords:** postoperative pain, fibula free flap, ropivacaine infiltration, local anesthetic toxicity

## Abstract

**Introduction:** Pain after cervicofacial cancer surgery with free flap reconstruction is both underestimated and undertreated. There is a rational for regional anesthesia at the flap harvest site, but few studies describe it. We assessed the influence of common peroneal nerve infiltration on pain and opioid consumption in patients having oropharyngeal cancer surgery with fibular free flap mandibular reconstruction. **Methods:** After institutional review board (IRB) approval and written informed consent, fifty-six patients were randomly allocated to perineural catheter with ropivacaine infiltration (ROPI) or systemic analgesia (CONTROL). In the ROPI group, an epidural catheter was placed by the surgeon before closure, and ropivacaine 0.2% 15 mL, followed by 4 mL/h during 48 h, was administered. The primary outcomes were pain scores and morphine consumption during the 48 h postoperative period. We also measured ropivacaine concentration at the end of infusion. Finally, we retrospectively assessed long-term pain up to 10 years using electronic medical charts. **Results:** Perineural infiltration of ropivacaine significantly reduced pain scores at the harvest site only at day 1, and did not influence overall postoperative opioid consumption. Ropivacaine assay showed a potentially toxic concentration in 50% of patients. Chronic pain was detected at the harvest site in only one patient (ROPI group), and was located in the cervical area in the case of disease progression. **Discussion:** Although the catheter was visually positioned by the surgeon, continuous ropivacaine infiltration of the common peroneal nerve did not significantly reduce postoperative pain, but induced a blood concentration close to the toxic threshold at day 2. Further studies considering other infiltration locations or other dosing schemes should be tested in this context, both to improve efficacy and reduce potential toxicity.

## 1. Introduction

Controlling postoperative pain after major cancer surgery is a daily goal for anesthesia teams and a major expectation for the patients.

Efficient postoperative analgesia is reported to positively affect not only the patient’s comfort, but many outcomes including mortality, surgical outcomes, hospital stays and costs [1]. It is also a key condition to prevent chronic pain.

Despite the combination of several classes of drugs, constant improved knowledge over years and anticipation, pain is still insufficiently relieved, especially in major cervicofacial cancer patients, both because it is underestimated by communication impairment after tracheotomy and because the needs for analgesia differ enormously between patients [2,3].

Moreover, after oropharyngeal cancer surgery, reconstruction is often necessary, and harvesting a disease-free bone or muscle flap might induce a second location of severe postoperative pain [4]. Traditional postoperative management of major cervicofacial surgery with free flap reconstruction is a combination of systemic drugs including opioids, non-steroidal anti-inflammatory drugs and other systemic analgesics such as nefopam and paracetamol.

When the flap is taken from a limb such as the fibula flap, additional regional analgesia can be proposed for the common peroneal nerve to provide a postoperative sensory block. This is expected to decrease pain at the harvest site and improve rehabilitation, although the benefit on opioid consumption is unpredictable, since regional analgesia on the limb has no influence on the pain at the cervical site. 

Few studies have looked at regional analgesia after free flap reconstructive cervicofacial surgery. Zhang, using a double block bolus [5], and Ferri et al., giving repeated boluses every 8 h through a perineural catheter [6], found a significant benefit on pain after fibula free flap. Conversely, Roof et al. observed no benefit of a constant infusion (ropivacaine 0.2% 6 mL/h, but on a very small group of patients (*n* = 8) [7].

However, as the pain at the harvest site is expected to last several days [4], there is a rational for continuous perineural infusion that should be further studied.

The purpose of this randomized controlled open study was first to describe the influence of a continuous local anesthetic perineural infiltration at the harvest site (common peroneal nerve) on early postoperative pain and opioid consumption after fibula free flap for mandibular reconstruction.

Subsequently, we assessed the risk of systemic ropivacaine toxicity by collecting blood samples for ropivacaine assay at the end of infusion on postoperative day 2, i.e., at the supposed maximal value of blood concentration. Finally, we retrospectively assessed possible long-term postoperative pain at the cervical and harvest site.

## 2. Methods

The protocol was approved by the ethical committee of Henri Mondor Hospital, Créteil France (CPPRBP # 04-006), and by our local hospital institutional review board (CSET # 03-1042) in April 2004. Patients scheduled for oropharyngeal cancer or post-cancer (radionecrosis) surgery with fibula free flap reconstruction were eligible. Written informed consent was obtained for each patient during anesthesia preoperative consultation.

Patients were then allocated by a computerized list of random numbers in a 1:1 ratio between ropivacaine regional analgesia of the harvest site associated with systemic analgesia (ROPI group) and systemic analgesia alone (CONTROL group).

Premedication was at the discretion of anesthesiologists. The medical team was aware of randomization, while the patients were not.

Anesthesia was started with remifentanil effect-site target-controlled infusion (Base Primea TCI pump, Minto pharmacokinetic model), propofol and atracurium for tracheal intubation, and maintained with remifentanil, inhalational anesthesia (desflurane or sevoflurane in a mixture of O_2_/N_2_O 50% each) and bolus injections of atracurium if needed.

Before skin closure, an epidural catheter with end-tip holes only was placed in the ROPI group next to the proximal end of the common peroneal nerve by the plastic surgeon and was fixed by a suture in the skin. After skin closure, 15 mL of ropivacaine 0.2% was injected through the catheter. No further control of the tip of the catheter was performed. This was followed by continuous infusion of ropivacaine 0.2% 4 mL/h initiated in a post-anesthesia care unit (PACU) and administered during 48 h (Ambi T^TM^ PCA pump).

Multimodal postoperative analgesia with other intravenous analgesics (paracetamol, tramadol, nefopam) and morphine 0.15–0.2 mg/kg was administered to all patients.

All patients had planned tracheotomy performed during surgery: postoperative pain assessment took into account this communication difficult, which was explained to the patients preoperatively (day-1).

In the PACU, intravenous (IV) morphine titration was started until visual analog scale < 30/100, and continued with patient-controlled analgesia (PCA) of morphine (no continuous infusion, bolus of 1 mg allowed every 5 min).

The following parameters were recorded: demographic characteristics, duration of anesthesia, remifentanil consumption, intraoperative morphine doses, PACU titration doses and postoperative morphine consumption.

Pain was assessed regularly using a 100 mm visual analog scale (from 0 to 100) both at the fibula site (harvest site) and cervical site.

One blood sample for ropivacaine assay was collected at postoperative hour 48 or at the time of catheter removal (if removed earlier) in 16 of the patients who were allocated to receive ropivacaine. Venous blood was sampled in heparinized tubes. The plasma separated by centrifugation was stored at −18 °C until assayed. Ropivacaine was measured using gas chromatography with a limit of quantification of less than 0.01 µg⋅mL^−1^. The intra- and inter-day coefficients of variation were 6 and 8% at 0.2 µg⋅mL^−1^ [8].

For late postoperative assessment, which was not part of the initial study, a second IRB approval was obtained to extract additional information on possible late postoperative pain (harvest and cervical site) from the electronic chart of patients in September 2020. Retrospective electronic medical chart review up to 10 years was performed. The presence of pain at the cervical and harvesting site and requirement of analgesic treatment was recorded 1, 3, 5 and 10 years after the reconstructive surgery.

## 3. Statistical Analysis

The sample size was chosen in order to decrease postop VAS at rest from 40 (usual value) to 30 mm in the ROPI group with a common standard deviation of 13 mm, with a bilateral risk error alpha = 0.05 and a power > 90%. For this purpose, 29 patients were necessary in each group.

We checked the normality of distribution of variables using the Shapiro–Wilk test. Continuous variables were compared with a Student’s *t*-test or a Mann–Whitney U test and expressed as the mean ± standard deviation or median (interquartile range, IQR), as appropriate. For categorical data, a chi-square test or Fisher’s exact test was used. *p* < 0.05 was considered as statistically significant. Statistical analysis was performed with Medcalc v15.4 statistical software (Ostend, Belgium).

## 4. Results

The study was conducted from June 2004 to November 2007. 

The flowchart is displayed in Figure 1. In each group, 29 patients were randomized. Then, 2 were excluded in the ROPI group because surgery was aborted for surgical concerns. Thus, 27 patients in the ROPI group and 29 patients in the CONTROL group were included in this modified post-randomization intention-to-treat analysis. 

No difference was noticed in demographic and intraoperative characteristics (Table 1) except for remifentanil consumption, which was significantly higher in the ROPI group.

Postoperative pain score at the harvest site was significantly less at post-op day 1 (H 28, Figure 2) only.

No significant difference was observed between groups in pain score at the cervical site (Figure 3).

Opioid consumption, which is a global consequence of pain at both sites, did not differ significantly between groups (Figure 4).

No adverse events related to the postoperative analgesic catheter and no complications at the harvest site were observed in any group.

Time to discharge from the hospital was not different between groups (Table 1).

The catheter was removed 48 after ropivacaine infusion initiation in 25 patients and earlier in the remaining two. In these patients, removal occurred at 12 and 20 after infusion initiation. Analgesic data were analyzed until the time of removal.

Ropivacaine blood concentration in 16 patients in the catheter group was 2.1 ± 1.1 µg/mL with a minimum 0.36 of and maximum of 3.96 µg/mL (Figure 5). Although 8 patients (50%) had a blood concentration above the threshold of 2.2 µg/mL, classically considered as high risk for systemic toxicity [9], no symptoms were observed in any patient in the ROPI group. Unbound ropivacaine concentration was not assayed. Postoperative protidemia (Day 1) was 49 ± 5 g/L, i.e., a mean loss of 20 g/L in comparison to preoperative values of (69 ± 5 g/L).

All patients had at least 1-year follow-up and one third of them had retrospective electronic assessment up to 10 years. In this long-term retrospective analysis, only one patient, in the ROPI group had moderate chronic pain sequelae in the fibula harvest site, which lasted up for 6 years. No other pain in the harvest site was reported.

Few patients developed chronic pain at the cervicofacial site due to initial surgery. However, some of them declared new cervicofacial pain developed later in relation to cancer evolution, new cervicofacial cancer or infection (Table 2).

This long-term follow-up was not planned in the initial randomized study; therefore, these data cannot be considered as the result of the initial study, but rather as retrospective observational results with an important percentage of lost follow-ups.

## 5. Discussion

This study shows that a continuous infiltration of ropivacaine 0.2%, administered next to the upper extremity of the common peroneal nerve for 48 h, only significantly reduced pain scores at the harvest site of a fibular free flap at day 1, without significant influence on pain scores at other time points, nor on cumulative morphine consumption. Ropivacaine assay performed in 16 patients revealed a high concentration of local anesthetic in 8 patients.

Among numerous free flap reconstructive options available after major cervicofacial surgery, we focused on the fibula because it is considered as the most efficient flap regarding aesthetic and functional rehabilitation (swallowing, eating and speaking) for mandibular reconstruction. It is often proposed in cancer surgery, may gather an homogeneous population and is eligible for limb regional analgesia. In addition, it is known for delayed healing and chronic pain, estimated between 2% and 60% of the patients [10,11,12], which both may be improved by optimal postoperative pain management. 

Therefore, there was a strong rational for regional infiltration at the harvest site. It should last at least for 24 h, but may not be necessary later than day 2, since pain markedly decreased after 48 h [5,7].

Among the few studies available, a double block (femoral + common fibular nerve with ropivacaine 0, 33%) decreased pain score until the 12th postoperative hour at rest and the 8th hour in movement [5]. Repeated chirocaine boluses every 8 h decreased pain score on average from 4 before reinjection to less than 1 after reinjection [6]. Conversely, ropivacaine continuous infiltration through a catheter placed by the surgeon was unable to significantly improve pain scores [7], even with a similar drug delivery protocol as in our study [13].

Several explanations may be proposed to explain the discrepancy between these results and ours.

First, infusing in front of the peroneal nerve may be too distal to cover the whole surgical harvest zone. Combining a femoral block or placing the catheter more proximal in the poplitea fossa in front of the sciatic nerve may improve the efficacy on postop pain. Additionally, in our study, the catheter was placed by the surgeon resident and its placement may have been imperfect. Preoperative block performed with ultrasound control by the anesthesiologist may cover a wider zone and be more reproducible.

Moreover, placing a catheter for postoperative reinjections induces a risk of catheter displacement after only a few hours, as observed by Marhofer [14], which may have happened to some of our patients. This does not contraindicate placing a catheter; the more distant from the surgical zone and articulation, the lesser would be the risk.

On the other hand, the doses given might have been inappropriate to achieve pain relief. Firstly, an initial 15 mL bolus, which we used in this study, is reported to be the most adapted dose before reaching a ceiling effect [13], while An infusion rate of 4 to 6 mL/h of ropivacaine 0.2% is the classically recommended dose for perineural block of limb or chest, and as a balance between underdosage and potentially toxic doses.

However, in this special localization, a more important diffusion volume or systemic absorption and increased local inflammation might explain the high blood concentrations observed.

Ferri et al. gave boluses of 20 mL every 8 h [6]; by doing so, they may have achieved a better local diffusion at higher injection pressure with less systemic resorption, as suspected in a recent meta-analysis [15]. This issue should be studied for every location of perineural block, since the balance between local efficacy and systemic absorption differs with the diffusion space of each block and the inflammatory state around it.

Besides its disappointing efficacy, the ropivacaine infiltration, as administered in our study, induced, at the end of infusion, venous blood concentration above the maximal usually tolerated threshold of 2.2 µg/mL in 50% of patients [9]. In 2 of them, it induced concentrations higher than 3.7 µg/mL, which have been described inducing seizures [16].

Another study infusing ropivacaine in the extra-pleural space after thoracotomy observed lower concentrations than in our study, despite higher doses (6 or 9 mL/h) [17]. Fortunately, no sign of toxicity was observed in our study, similar to a previous work with a similar range of concentrations [18].

Toxicity is correlated not to the total, but to the unbound fraction, which was not recorded in our study. Hypoprotidemia due to preoperative denutrition or to intraoperative hemodilution may worsen the risk. However, usually this hypoprotidemia involves mainly albumin, whereas Ropivacaine binds mainly to alpha-1-acid glycoprotein (AAG), which increases after trauma or inflammation [19], Consequently, the major inflammation state observed in cervicofacial cancer surgery may have protected the patients from unbound ropivacaine toxicity by decreasing the unbound fraction, especially after prolonged infusion, as shown by Blumenthal [20].

In recent studies on cervicofacial surgery, pain is maximum in the first postoperative hours or day, but 50% of patients already have no pain at discharge [2]. This result is consistent with ours, with only one patient complaining of chronic pain at the harvest site, which resolved after several years.

Finally, the influence on chronic pain did not appear as relevant as expected from the literature [10,21]. Some reviews are based on old original papers where multimodal analgesia including anti-NMDA agents such as N_2_O, nefopam or ketamine were not used, whereas they were used in our study.

Our study had a few shortcomings; firstly the sample size has a statistical power insufficient to detect differences in postoperative pain scores, morphine requirements or potential chronic pain at the harvest site. Average values were similar to the assumption used for the number of subjects calculation, but interindividual variability was underestimated. In other words, some patients have very high levels of pain and analgesic requirements, and risk factors for these outliers are not easy to establish [2].

This result supports the recommendation to regularly assess the pain scores of patients after this surgery, and to adapt analgesia protocol to the individual needs of outliers [5].

The higher remifentanil rate of infusion may be due to stronger pain in the ROPI group, which may have offset a potential benefit of regional anesthesia. However, it may also be due to a random effect in this rather small groups of patients.

In this study, the epidural catheter placed by the surgeon near the proximal side of the common peroneal nerve can partly be assimilated to wound infiltration, despite the fact that the epidural catheter has holes only in its terminal end. Nevertheless, the complexity of this type of surgery may yield heterogeneous pain results as in our study; we reference regional techniques such as popliteal ultrasound guided block with intermittent reinjection, which might be more effective and more reproducible [22].

Other shortcomings were that allocation to one or the other group was unblinded to professionals, which could have yielded bias in harvest site pain-score assessment, but not in morphine PCA consumption. Finally, long-term assessment was retrospectively performed through electronic chart database consultation, which could provide some bias or missing data.

## 6. Conclusions

Ropivacaine continuous infiltration of 4 mL/h for 48 h in front of the common peroneal nerve cannot be recommended for analgesia after fibular free flap because of both insufficient analgesic efficacy and high, potentially toxic, blood concentrations.

Future research and clinical practice should consider more proximal blocks and/or different drugs delivery schemes including repeated boluses and continuous search of the lower dilution efficient for each type of block, and drug assay on a few patients for every new block or scheme tested.

A huge variability in pain and analgesic requirements between patients supports the recommendation of repeated pain assessment and individualized analgesic drug adjustments, especially in the first days after surgery.

## Figures and Tables

**Figure 1 jcm-11-06384-f001:**
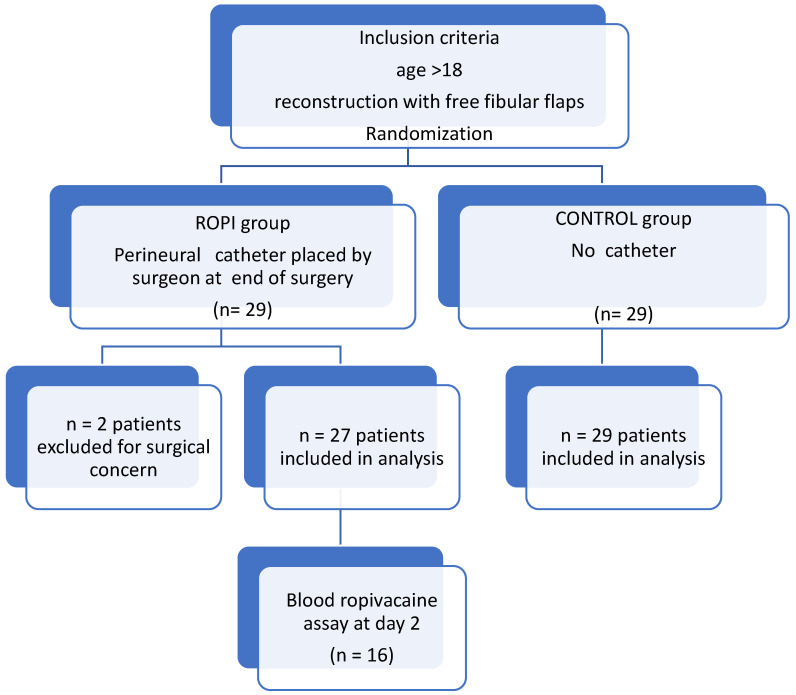
Study flow chart.

**Figure 2 jcm-11-06384-f002:**
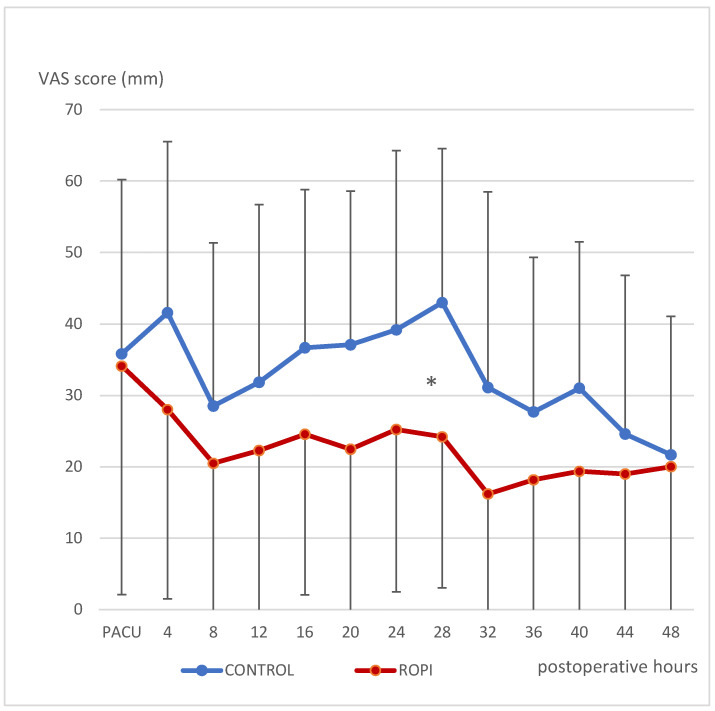
Pain scores at rest (VAS mm) at fibula harvest site. Mean and SD vs. postoperative time (hours) starting in PACU. * *p* < 0.05.

**Figure 3 jcm-11-06384-f003:**
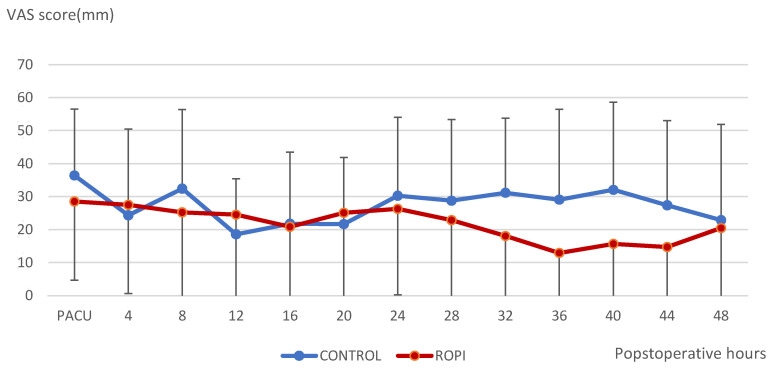
Pain scores at the cervical site at rest; mean and SD vs. postoperative time (hours) starting in PACU.

**Figure 4 jcm-11-06384-f004:**
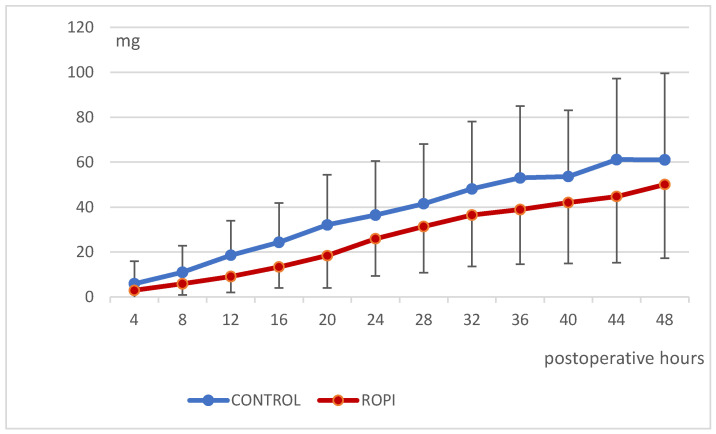
Cumulative morphine consumption (Mean and SD) until 48 h postoperative.

**Figure 5 jcm-11-06384-f005:**
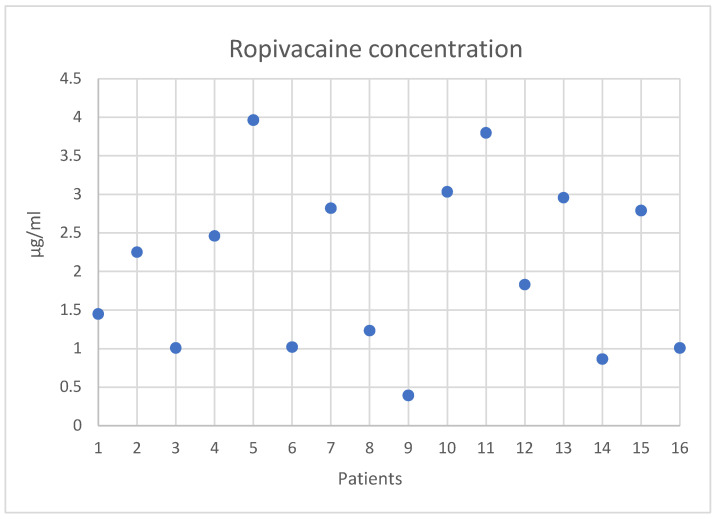
Individual values of ropivacaine blood concentration at the end of infusion.

**Table 1 jcm-11-06384-t001:** Demographic and perioperative data (M ± SD).

	ROPI (*n* = 27)	CONTROL (*n*= 29)	*p* Value
Age (y)	53 ± 15	53 ± 15	NS
Weight (kg)	69 ± 16	64 ± 13	NS
Height (cm)	169 ± 7	169 ± 8	NS
Gender male/female (*n*)	21/6	19/10	NS
Preoperative analgesics (number of patients) Opioid Non-opioid	6/27 3 3	8/29 8 0	NS
Remifentanil average rate (µg⋅kg^−1^⋅min^−1^)	0.092 ± 0.033	0.074 ± 0.031	0.048 *
Duration of anesthesia (min)	557 ± 58	570 ± 58	NS
Morphine cumulative consumption D0-D2 (mg)	51 ± 32	61 ± 38	NS
Length of stay (day)	22 ± 12	23 ± 10	NS

* *p* < 0.05.

**Table 2 jcm-11-06384-t002:** Chronic pain patients requiring medication (number of patients/number of patients assessed).

Delay from Surgery	ROPI (*n* = 27)	CONTROL (*n*= 29)	*p* Value
1 year	3/27	5/29	0.7
3 years	4/17	5/20	1
5 years	3/15	4/14	0.6
10 years	1/10	1/9	1

## Data Availability

Data are available on demand to corresponding author.

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
