# Peer review of "Continuous Ropivacaine Peroneal Nerve Infiltration for Fibula Free Flap in Cervicofacial Cancer Surgery: A Randomized Controlled Study"

_jcm, 2022, doi:10.3390/jcm11216384_

Round 1
Reviewer 1 Report
Abstract
- Please describe IRB (abstract) and ROPI as they appear for the first time in the text.
- 10 years rather than 10 year. (plural).
Key words: English correction: “posroerative pain” first key word.
METHODS: Authors should cite the period of recruitment of patients scheduled for the surgery in which the study was conducted. Taking into consideration that study was performed in 24 months and that the follow up was 10 years, then it must have been sarted at 2010 or before. This should be in line with ethical approval and clinical trial registration protocols.
- There is no mention to registration of the study in a clinical trial registration service. Authors should double check this item to make sure this is in line with the journal’s requirements and guidelines for randomized clinical trials.
Authors should describe the allocation rate (I presume 1:1) right on methods section.
Authors should describe the randomization protocol that was used.
Line 118. There is a closing parenthesis without a corresponding opening parenthesis
Line 124. “The study was conducted during 24 consecutive months”. Is 2 or 10 years?
data for chronic pain assessment was based on “Presence of pain requiring analgesic treatment” This is the criteria that authors used to monitor pain at 1, 3, 5 and 10 years. As exist many reasons that may correlate to analgesic treatment of pain (not only the surgery), the data on chronic pain is quite elusive and even if there were statistical diference, I feel skeptical that it may correlate with the perioperative ROPI infusion.
Line 134. but due to huge standard deviations, it reached statistical significance only at 134 postop day 1.
I strongly disagree with such affirmation. Altough statistical significance can be affected by the standard deviation of the sample, it is totally elusive and not assertive to stablish an straight cause-consequence relationship between them. Alternatively, authors should describe their data with a narrative and more neutral point of view.
Figures 2, 3 and 4. There is critical missing information. Both X and Y axis have no labels. It MUST have. Eg. X (in hours?) Y (VAS?) (mg of opioid?? in fig. 4), etc.
Author Response
Thank you for your constructive comments
Please describe IRB (abstract) and ROPI as they appear for the first time in the text.
We described the two items as requested (institutional review board and ropivacaine in the abstract
- 10 years rather than 10 year. (plural). done
Key words: English correction: “posroerative pain” first key word.
done
METHODS: Authors should cite the period of recruitment of patients scheduled for the surgery in which the study was conducted. Taking into consideration that study was performed in 24 months and that the follow up was 10 years, then it must have been sarted at 2010 or before. This should be in line with ethical approval and clinical trial registration protocols.
- There is no mention to registration of the study in a clinical trial registration service. Authors should double check this item to make sure this is in line with the journal’s requirements and guidelines for randomized clinical trials.
Thank you for your suggestion , this study was initially approved by the ethical committee in early 2004 , the first patient was enrolled in June 2004 , at this time there were no obligation to submit the study to the clinical trial gov , or the equivalent body in Europe which was created later.The initial approval was for 24 months , but because of a change in the investigator team additional ethical committee approval was obtained on May 2005 to prolong the study. All certificates and approvals are available for the editorial board upon request
For the late postoperative pain outcomes additional IRB approval was obtained on september 2nd 2020 as part of a retrospective electronic chart analysis.
Authors should describe the allocation rate (I presume 1:1) right on methods section.
Authors should describe the randomization protocol that was used.
We used a computerized list of random numbers
Thank you for your suggestion we made changes as requested as follows
Patients were allocated by a computerized list of random numbers in a 1:1 ratio between ropivacaine regional analgesia of the harvest site (ROPI group) and systemic analgesia alone (CONTROL group).
Line 118. There is a closing parenthesis without a corresponding opening parenthesis
Thank you this was corrected
Line 124. “The study was conducted during 24 consecutive months”. Is 2 or 10 years? patients were recruited for two years , the followup was for 10 years , this latter part was not part of the initial study , however additional authorization was obtained from our IRB for the electronic chart followup of the patients
data for chronic pain assessment was based on “Presence of pain requiring analgesic treatment” This is the criteria that authors used to monitor pain at 1, 3, 5 and 10 years. As exist many reasons that may correlate to analgesic treatment of pain (not only the surgery), the data on chronic pain is quite elusive and even if there were statistical difference, I feel skeptical that it may correlate with the perioperative ROPI infusion.
Thank you for your comment , we agree that the data on chronic pain do not correlate with the initial infiltration protocol, indeed they are not perfect especially as the initial study was not designed for this , however these data exist and we consider them as additional data bringing some more information especially that many of these patients suffer chronic secondary pain , in the cervical site often in relation to a secondary or the cervicofacial cancer we agree again that these objectives were not initially planned but they deserve to be communicated as observationnal data in relation to this type of procedure which are not prolific.
Line 134. but due to huge standard deviations, it reached statistical significance only at 134 postop day 1.
I strongly disagree with such affirmation. Altough statistical significance can be affected by the standard deviation of the sample, it is totally elusive and not assertive to stablish an straight cause-consequence relationship between them. Alternatively, authors should describe their data with a narrative and more neutral point of view.
Thank you for your suggestion , we changed this paragraph as follows:
Postoperative pain score at the harvest site was significant at postop day 1 (H 28, ) only.(figure 2)
No significant difference was observed between groups in pain score at cervical site (figure 3),
Figures 2, 3 and 4. There is critical missing information. Both X and Y axis have no labels. It MUST have. Eg. X (in hours?) Y (VAS?) (mg of opioid?? in fig. 4), etc.
We totally reedited figure 3-4 and 5 and we thank you for your constructive comment
Reviewer 2 Report
This manuscript is very well written and presents a study that is well executed, but possibly suffering from some suboptimal planning that reflects in the somewhat disappointing results. Nevertheless, it is a contribution to our knowledge and step towards further studies of this challenging scenario – major surgery with significant donor site pain.
Abstract (and Discussion) – talking about rationale for regional anesthesia, but the method described is closer to local anesthetic infiltration, not regional anesthesia (e.g. sciatic nerve block) since the catheter is laid inside the surgical wound and its position subject to significant change
Questions arising from the results:
- Did the authors compare the (difference in the) pain scores in the peroneal and the cervical sites in each patient?
o It seems from the composite scores that the two surgical sites were similar within each group and possibly equally affected by the ropivacaine infusion
o If the above proves to be true, one could speculate that the system local anesthetic effect was dominating the benefits? Please, explore and explain
- The reported ropivacaine plasma levels are very surprisingly high for the modest bolus and infusion rate? How do you explain that?
o Was more ropivacaine used for surgical infiltration at the cervical site and how much?
- What attempt was made to confirm correct catheter position after the end of surgery? Sensory testing to confirm no secondary failure?
- Page 7 line 175 – please, consider rephrasing “upper extremity of the common peroneal nerve” to avoid misinterpretation – suggest “proximal aspect of…”
- Discussion (and Abstract) – the method described is closer to local anesthetic infiltration, not regional anesthesia (e.g. sciatic nerve block) since the catheter is laid inside the surgical wound and its position subject to significant change; this might be another explanation of high local anesthetic plasma levels (and low analgesia efficacy at the donor site)
o Recommend including another paragraph on the shortcomings of surgically placed nerve blocks…
- The Discussion is almost perfect, possibly benefiting from previous revisions. It reflects adequately the problems and shortcomings and should need only some edition and small additions
Author Response
We thank the reviewer for his constructive suggestions and comments
Questions arising from the results:
- Did the authors compare the (difference in the) pain scores in the peroneal and the cervical sites in each patient?
No We did not compare the difference in pain scores between peroneal and cervical site for each patient
o It seems from the composite scores that the two surgical sites were similar within each group and possibly equally affected by the ropivacaine infusion
We did not infiltrate the cervical site , we added a sentence in the method section and we apologize if this caused confusion , we guess due to an error in editing the figures the reviewer might been induced to misinterpretation , we totally reedited the figures and the pain scores in the cervical site and the harvest site are different
o If the above proves to be true, one could speculate that the system local anesthetic effect was dominating the benefits? Please, explore and explain
- The reported ropivacaine plasma levels are very surprisingly high for the modest bolus and infusion rate? How do you explain that?
the plasma concentration in some patients were high may be to the bad nutritionnal status of our cancer patients and may be to a postoperative dilution and a decrease in proteins albuminemia or due our delivery system
o Was more ropivacaine used for surgical infiltration at the cervical site and how much?
As we previously answered above no ropivacaine was infiltrated in the cervical site
- What attempt was made to confirm correct catheter position after the end of surgery? Sensory testing to confirm no secondary failure?The infitration catheter was placed by the plastic surgeon under direct vision and was fixed with a suture to the skin, no further checking was performed.
The catheter we used was in fact an epidural catheter with wholes at the end tip only (3) , at the time of the study infitration catheters were not commercialized. therefore shothhcomings related to infiltration catheter may not be totally applicable to the epidural catheter placed nearby common peroneal nerve.
Before skin closure, an epidural catheter with end-tip holes only was placed in the ROPI group
- Page 7 line 175 – please, consider rephrasing “upper extremity of the common peroneal nerve” to avoid misinterpretation – suggest “proximal aspect of…”
This sentence has been rephrased thank you for your suggestion
Before skin closure, an epidural catheter was placed in the ROPI group next to the proximal end of the peroneal nerve by the plastic surgeon and was fixed by a suture in the skin.
- Discussion (and Abstract) – the method described is closer to local anesthetic infiltration, not regional anesthesia (e.g. sciatic nerve block) since the catheter is laid inside the surgical wound and its position subject to significant change; this might be another explanation of high local anesthetic plasma levels (and low analgesia efficacy at the donor site)
, We agree that method is rather surgical site infiltration although the epidural catheter had hole in the terminal extremity , we tried to use this term whenever we used regional block , in addition we also changed the title accordingly.
o Recommend including another paragraph on the shortcomings of surgically placed nerve blocks…
Thank you for your suggestion we addeded the following paragraph in the discussion section
In this study the epidural catheter placed by the surgeon near the proximal side of the common peroneal nerve can partly be assimilated to wound infiltration despite that the epidural catheter has wholes only in its terminal end. Nevertheless the complexity of this type of surgery may yield heterogeneous pain results as our study, we believe reference regional techniques such as popliteal ultrasound guided block might have been more effective (22).
- The Discussion is almost perfect, possibly benefiting from previous revisions. It reflects adequately the problems and shortcomings and should need only some edition
Thank you so much in fact this is the first revision and this paper was not submitted elsewhere before , we reviewed the discussion and made some minor changes and took acount remarkes by other reviewers and yourself
Reviewer 3 Report
This “randomized controlled” study investigated the analgesic effect of perineural ropivacaine infiltration (0.2% 15 ml bolus and then 4 ml/h infusion for postoperative 48 h) for fibula free flap surgery and the blood ropivacaine concentrations after 2-day continuous infusion. Some major revisions and general English wording are suggested.
1. P2, L62 and P3, L83–84: Please confirm the target location of the epidural catheter tip at common peroneal nerve, lateral sural cutaneous nerves, or superficial peroneal nerve by surgeon during operation, to provide postoperative sensory block for fibula free flap surgery.
2. P2, L69–73: Please state the conducted year and duration of this study. Was the 10-year follow-up included in the initial IRB protocol of this RCT?
3. P2, L74–75: Please comply with the CONSORT 2010 checklist for RCT and describe the sequence generation, allocation concealment, and implementation of randomization. Otherwise, this is only an observational study with a great loss to follow-up, which could not determine the validity of outcome analysis and chronic pain rate.
4. L3, P112–115: According to your sample size calculation, 29 patients were necessary in each group. However, more participants should be enrolled in the protocol for possible drop-out or loss to follow-up. This leads to the critical limitation stated in page 9, L245, which may contribute to the insignificant lower VASs and morphine consumption in the ropivacaine group.
5. P3, L116: Please confirm the outcome analysis with intention to treat or modified intention to treat with post-randomization exclusion of 2 patients.
6. P4, Figure 1: Please delete the “Long-term assessment of pain up to 10 years”, if you did not include this outcome in the initial IRB protocol for the RCT.
7. P4, L132: Please move L148–155 and Table 1 to L133, before pain score section.
8. P6, L161: Please provide preoperative values of protidemia to calculate the exact postoperative loss. Otherwise, it’s only an “estimated” loss by normal preoperative values.
9. P6, L164–172: In Table 2, the comparison of chronic pain after one year between two groups has a very low “Level of Evidence” with two-thirds of loss to follow-up after 10 years in both groups. This part should be omitted in a genuine RCT with a strict protocol.
10. P8, L190–209: Regarding the sensory nerve branch and ropivacaine dose for pain relief, please refer to: Christiansen CB, et al. Volume of ropivacaine 0.2% and common peroneal nerve block duration: a randomised, double-blind cohort trial in healthy volunteers. Anaesthesia. 2018 Nov;73(11):1361–1367. doi: 10.1111/anae.14400.
11. Please revise the following words:
P1, L21: 10 years;
P1, L31: postoperative;
P2, L48: non-steroidal anti-inflammatory drugs;
P4, Figure 1: at days 2;
P6, Table 1: Preoperative analgesics (no of patients)
P7, Figure 5: Ropivacaine;
P7, L163: Individual.
Author Response
This “randomized controlled” study investigated the analgesic effect of perineural ropivacaine infiltration (0.2% 15 ml bolus and then 4 ml/h infusion for postoperative 48 h) for fibula free flap surgery and the blood ropivacaine concentrations after 2-day continuous infusion. Some major revisions and general English wording are suggested.
- P2, L62 and P3, L83–84: Please confirm the target location of the epidural catheter tip at common peroneal nerve, lateral sural cutaneous nerves, or superficial peroneal nerve by surgeon during operation, to provide postoperative sensory block for fibula free flap surgery.
Thank you for your suggestion we made suggested changes
Before skin closure, an epidural catheter with end-tip holes only was placed in the ROPI group next to the proximal end of the common peroneal nerve by the plastic surgeon and was fixed by a suture in the skin
2. P2, L69–73: Please state the conducted year and duration of this study. Was the 10-year follow-up included in the initial IRB protocol of this RCT?
The initial randomized study was performed from june 2004 until november 2007 this is now stated in the results section
A second IRB approval was obtained to extract additional information on late postoperative pain from the electronic chart of patients in September 2020.
you are right this part was not included in the initial randomized study .
P3, L83–84:
When the flap is taken from a limb such as fibula flap, additional regional analgesia can be proposed for common peroneal nerve to provide postoperative sensory block
Although the catheter was visually positioned by the surgeon, continuous ropivacaine infiltration of the common peroneal nerve did not reduce significantly postoperative pain but induced blood concentration closed to toxic threshold at day 2
- P2, L74–75: Please comply with the CONSORT 2010 checklist for RCT and describe the sequence generation, allocation concealment, and implementation of randomization. Otherwise, this is only an observational study with a great loss to follow-up, which could not determine the validity of outcome analysis and chronic pain rate.
Thank you for your suggestions , We modified figure 1; the flowchart of the study , we deleted the frame which included the longterm follow up , as the first part of the study which was a randomized open label study ,
the long term assessement was a retrospective electronic chart review ; we totally agree that these data should be interpretated very cautiously and were not part of the initial randomised study , in fact only this part of this paper can be considered as an observational study as the reviewer very righfully suggested however these data are additional true informations available for the reader and may not be ignored , we now mentionned it several times in the method and the discussion section.
The long term follow up was not planned in the initial randomized study , therefore these data cannot be considered as the result of the initial study, however since they are available for the reader ;these specific results should be considered very cautiously as an additional retrospective observational study. with important percentage of lost to follow up.
- L3, P112–115: According to your sample size calculation, 29 patients were necessary in each group. However, more participants should be enrolled in the protocol for possible drop-out or loss to follow-up. This leads to the critical limitation stated in page 9, L245, which may contribute to the insignificant lower VASs and morphine consumption in the ropivacaine group.
We agree that the sample size may not be adequate , indeed we expected the infiltration to be much more effective unfortunately we cannnot recruit more patients
- P3, L116: Please confirm the outcome analysis with intention to treat or modified intention to treat with post-randomization exclusion of 2 patients.
The flowchart is displayed in figure 1. In each group, 29 patients were randomized. Then, 2 were excluded in the ROPI group because surgery aborted for surgical concerns. Thus, 27 patients in the ROPI group and 29 patients in the CONTROL group were included in the this modified post randomization intention to treat analysis.
- P4, Figure 1: Please delete the “Long-term assessment of pain up to 10 years”, if you did not include this outcome in the initial IRB protocol for the RCT.
Thank you for your suggestion we deleted the longterm assessment frame in figure 1as it was not in the initial IRB approval thank you for your suggestion
- P4, L132: Please move L148–155 and Table 1 to L133, before pain score section.
Done thank you for your suggestion
- P6, L161:Please provide preoperative values of protidemia to calculate the exact postoperative loss. Otherwise, it’s only an “estimated” loss by normal preoperative values.
Postoperative protidemia (Day 1) was 49± 5 g/L i.e. a loss of 20 g/L in comparison to preoperative values of (69. ± 5 g/L).
- P6, L164–172: In Table 2, the comparison of chronic pain after one year between two groups has a very low “Level of Evidence” with two-thirds of loss to follow-up after 10 years in both groups. This part should be omitted in a genuine RCT with a strict protocol.
We fully agree that this part was not included in the RCT , however we believe these informations may be useful for some readers, therefore we clearly specified that these information were supplemental observational retrospective data.
- P8, L190–209: Regarding the sensory nerve branch and ropivacaine dose for pain relief, please refer to: Christiansen CB, et al. Volume of ropivacaine 0.2% and common peroneal nerve block duration: a randomised, double-blind cohort trial in healthy volunteers. Anaesthesia. 2018 Nov;73(11):1361–1367. doi: 10.1111/anae.14400.
Thank you for your suggestion , we have added additionnal comments and refereed to this reference in the text thank you
In addition one may suggest the doses given might have been inappropriate to achieve pain relief. Firstly an initial 15 ml bolus which we used in this study is reported to be the most adapted dose before having a ceiling effect (13). Secondly, infusion rate of 4 to 6 ml/h of ropivacaine 0,2% is classically recommended dose for perineural block of limb or chest and is usually sufficient as a balance between underdosage and potentially toxic doses.
- Please revise the following words:
P1, L21: 10 years; done
P1, L31: postoperative; done
P2, L48: non-steroidal anti-inflammatory drugs; done
P4, Figure 1: at days 2; done
P6, Table 1: Preoperative analgesics (no of patients) done
P7, Figure 5: Ropivacaine; done
P7, L163: Individual. done